# Design of Optoelectronic Tracking Platform Driven by Ultrasonic Motor with a Novel Limiter

**DOI:** 10.3390/mi14112067

**Published:** 2023-11-07

**Authors:** Yongjin Liang, Song Pan, Lei Chen

**Affiliations:** 1College of Aerospace Engineering, Nanjing University of Aeronautics and Astronautics, Nanjing 210016, China; liang_jin@nuaa.edu.cn (Y.L.); iwannabeaseal@nuaa.edu.cn (L.C.); 2State Key Laboratory of Mechanics and Control for Aerospace Structures, Nanjing 210016, China

**Keywords:** optoelectronic tracking platform, structural design, ultrasonic motor, position control

## Abstract

A high-performance servo control system is the basis for realizing high-precision photoelectric tracking. With high position resolution and power-off self-locking, ultrasonic motors have a wide range of applications for high-precision positioning control. An optoelectronic tracking platform driven by two ultrasonic motors is proposed in this study. The shaft structure of the tracking platform is designed and modeled. The shaft structure is simplified, and a dynamic model is established to analyze the motion characteristics. The parameters of the limit mechanism are optimized based on the analysis. The shaft structure is built to verify the response characteristics of the tracking platform at different velocities. The results show that the proposed design can fully utilize the self-locking of ultrasonic motors for rapid automatic alignment of the axis system. The maximum response time is less than 55 ms. When the operating velocity is less than 70°/s, the positioning error is less than 0.055°, and the lower the speed, the smaller the positioning error.

## 1. Introduction

An optoelectronic tracking system is a sophisticated, integrated system that amalgamates optical technology, image processing techniques, precision mechanical engineering, and advanced control systems. It enables the rapid detection of targets and precise tracking and targeting, finding extensive applications across a spectrum of engineering and technical domains, including target tracking, precision aiming, image-guided navigation, aerial photography, and battlefield reconnaissance [1,2,3,4,5]. For example, in weapon systems, optoelectronic tracking platforms are used as the foundation for tracking, aiming, and engagement in missile systems, modern artillery turret aiming, radar antenna automatic aiming, and laser emitter tracking and aiming [2,3].

In situations such as aviation reconnaissance observation equipment, space target surveillance devices, infrared cameras, space telescopes, and underwater sonar detection for deep-sea operations, optoelectronic tracking platforms also play a crucial role [5,6,7,8,9]. These tracking platforms can be mounted on fixed platforms, yet their adaptability for deployment on mobile platforms such as naval vessels, reconnaissance vehicles, and satellites is equally critical [1,3,10]. These necessitate improvements in the responsiveness, accuracy, portability, and environmental adaptability of optoelectronic tracking platforms [11,12,13,14].

Electromagnetic motors are applied to most rotating devices, generate low torque, and are energy efficient in high-speed ranges, as shown in Table 1 [15,16]. When the motor is set in a low speed and small rotating platform, a gearbox is required to maintain a high driving torque at low speed. However, the gearbox makes it difficult to miniaturize mechanical apparatuses and to achieve high accuracy due to backlash. At present, the optoelectronic tracking platform is usually driven by an electromagnetic motor and combined with various intelligent control algorithms to achieve higher precision positioning. Wu proposed a joint active disturbance rejection control and sliding mode control method, which can track the target by electromagnetic motors with 0.1 mrad accuracy [17]. Hu proposed a disturbance rejection controller with friction compensation to improve the target tracking ability and anti-disturbance performance by electromagnetic motors, and the angle error is 0.0542° [18]. In order to achieve higher torque, lower rotation speed, and self-locking function, the above platforms often need to be combined with a worm reducer, which inevitably increases the complexity of the driving mechanism and has problems of long positioning time, poor electromagnetic compatibility, large volume, and small positioning stiffness [13,14,17,18,19,20]. Therefore, it is necessary to adopt a new type of motor to improve the situation. As a prime mover, a rotating traveling wave ultrasonic motor (USM) offers low velocity, friction drive, self-locking, high-power density, high-position resolution, high positioning stiffness, and excellent electromagnetic compatibility [15,21,22,23,24,25,26,27,28]. These unique characteristics have always attracted attention in the fields of aviation, medical care, robots, and precision driving [29,30,31,32,33]. The USM is suitable for a direct-drive mechanism that makes it possible to solve several problems because it generates high torque and is energy efficient in the low-speed range. Due to the feature of high holding torque caused by frictional force, it is preferable that the USM is used as an actuator in specific devices that require miniaturization and low self-locking. Consequently, it holds great promise for applications in optoelectronic tracking platforms.

However, due to the friction-driven nature of ultrasonic motors, they generate a sudden and substantial braking torque when the motor stops. When the platform abruptly halts or changes direction at certain speeds, it can lead to significant vibrations throughout the tracking platform, potentially causing damage to optoelectronic equipment and precision shaft systems. Additionally, for systems with high inertia, the platform can exhibit overshooting during rapid positioning, resulting in oscillations around the equilibrium position. Zhang developed a special test turntable for a particular type of indexer based on USM. Although the positioning accuracy is very high, there is a significant overshoot during the positioning process, which greatly prolongs the positioning time of the platform [34]. To solve the above problems, a novel tracking platform driven by the USMs is proposed. On the platform, an angle limiter for USM can achieve automatic alignment of the axis system after USM is power-off and achieve the reduction of motor overshoot as a means of improving the position servo performance.

The remainder of this paper is organized as follows. Section 2 presents the structure design and modeling of the tracking platform, including support of the shaft, USM installation, and a novel mechanical angle limiter. Section 3 simplifies the limit mechanism of the axis system and establishes a dynamics model to analyze its motion characteristics. Section 4 presents an experimental analysis of the positioning characteristics of the tracking platform. Section 5 presents a summary and conclusions.

## 2. Structure of Tracking Platform

### 2.1. Main Structure

The presented optoelectronic tracking platform will be equipped with an optical camera and a laser ranging module. It will be applied to aerial and vehicular platforms to achieve sea and ground target detection, search, automatic identification, lock, and target tracking. Traditional optoelectronic tracking platforms often use electromagnetic motors combined with worm gear reducers to achieve higher holding torque and self-locking functions for typhoon resistance. However, this inevitably increases the platform’s volume and weight. Due to their principle of friction drive, USMs can directly output low-speed and high torque and achieve self-locking functionality [19].

The tracking platform mainly includes the design of azimuth- and pitch-axis systems. Considering the weight of the load, the accuracy of the shaft, the technology, and economics, the pitch- and azimuth axis are made of 304 stainless steel hollow structures for the wire to pass through. Most of the remaining materials are aluminum alloy 6061. The pitch- and azimuth-axis systems are driven by a USM, and two encoders are used for the position sense, as shown in Figure 1a. The USM and high-precision encoder form a closed-loop feedback system to output precise angular displacement. The technical parameters are shown in Table 2.

The left and right halves of the pitch shaft are connected by a load ring. To ensure rotation accuracy, the pitch shaft is double-end-supported using four P4 bearings, and two angular contact ball bearings are mounted back-to-back in pairs with bearing clearance adjusted by a bearing end cover, as shown in Figure 1b. The left half is equipped with a USM and a limit mechanism. The right half is equipped with a 21-bit absolute encoder. The load ring is used to mount the optoelectronic module. The azimuth shaft is single-end-supported and installed on a stainless-steel shaft sleeve using a pair of angular contact bearings back-to-back. The 21-bit absolute encoder, USM drive unit, and limit mechanism are installed under the shaft sleeve.

With the installation method, both axis systems have enough anti-overturning torque and axis-system stiffness, which can effectively improve the rotary motion accuracy and extend the life of the precision rotary table.

### 2.2. Ultrasonic Motor Drive Unit

USM has no conventional motor windings or magnetic circuits. It utilizes the vibration of the elastic body (stator) in the ultrasonic frequency band and the reverse piezoelectric effect of piezoelectric materials. The mechanical movement and torque are obtained by means of the frictional contact force between the stator and rotor [19], as shown in Figure 2b.

The structure of the traditional traveling wave USM is shown in Figure 2a. Besides the stator and rotor of the motor, there are some auxiliary structures to maintain the pressure and coaxiality between the stator and rotor. For lightweight design, both axes are driven by hollow USMs, and the stator and rotor of the USM are directly installed on the shaft. In addition, the installation of the USM on the pitch-axis system is different from the azimuth-axis system. The USM rotor on the pitch-axis system is fixed on the shoulder by a support and drives the pitch shaft to rotate, as shown in Figure 3a. There is a bearing inside the stator fixed support; the outer ring of the bearing interference fits the support, and the inner ring clearance fits the pitch shaft. Thus, preload pressure between the stator and rotor can be provided by an elastic pad and a lock nut on the outside of the bearing.

The stator is fixed on the azimuth shaft shoulder by a stator support and drives the azimuth shaft to rotate, as shown in Figure 3b. The azimuth axis has a hole for wire routing to the encoder and stator, such that the stator and encoder allow for 360° rotations. The pre-pressure between the stator and the rotor is applied in the same way as in the pitch-axis system.

### 2.3. Mechanical Angle Limiter

The internal structure of the limiter is shown in Figure 4a. Two steel columns are tightly held against the limit holder under elasticity. On the pitch- or azimuth shaft, the limit holder is fixed on the rotor or stator support, the limit seat is fixed on the base, two springs and two steel columns are inserted into the through hole of the limit seat, and the spring pre-pressure is adjusted by the screws at both ends, as shown in Figure 4b,c.

The traditional positioning requires the ultrasonic motor to slow down gradually and finally approach the desired position. By the limiter, the ultrasonic motor can cut off power directly at the needed position when the shaft has a certain speed. When the USM is turned off, due to the USM’s self-locking characteristics, the shaft, stator, rotor, and limit holder rotate together and then return to the original position under the action of the limiter so that the positioning time is very short. When the USM starts, the stator and rotor rotate relative to each other; one is restricted by the limit mechanism, and the other drives the shaft rotation. Under the action of the limit mechanism, the impact between the stator and rotor can be reduced to achieve a smooth start. At the end of the movement, the shaft is automatically aligned by the limit mechanism.

## 3. Shaft Motion Characteristics

### Dynamic Characteristics

Figure 5 shows the 3D model of the limiter structure and a simple model. When the USM suddenly stops from a state of motion, the stator and rotor are locked; thereafter, the motion of the axis system can be regarded as a damped single-degree-of-freedom vibration system. The pitch shaft, stator, rotor, support, and limit holder can be regarded as a rigid element; the spring in the limit seat is an elastic element. To more easily understand the mechanical relationship between the shaft and the limit mechanism, the abstract model represents the small rotations of the shaft, stator, rotor, support, and limit holder as flat movements.

Let the equivalent mass of the pitch shaft, stator, USM rotor, support, and limit holder be *M*. The two springs are of the same type and have the same coefficient of elasticity k. In the equilibrium position, the springs are both compressed by *u*. When *M* moves Δ*u* to the right from the equilibrium position, the two springs together have a force on *M*:(1)F=k(u+Δu)+k(−u+Δu)=2kΔu
when *M* moves, it will receive the joint action of two springs. Thus, this structure can effectively increase the elasticity and compensate for the lack of rigidity of the small spring.

There is friction between the components, which constitutes frictional damping. Alternating stress within the material generates friction, which constitutes structural damping. As it is a nonlinear damping model with difficulties in analytical calculation, the energy dissipation effect of different types of damping in the actual analysis can be represented as viscous damping.

After the USM is powered off, the stator and rotor are locked immediately. Due to the inertia of the shaft system, the shaft system will continue to move forward and then gradually stop and return to the power-off position under the action of springs and damping in the limit mechanism. A force analysis of the axis system was performed, and kinetic equations were established by the theorem of momentum:(2)$$J\theta^{\prime \prime }+2cl^{2}\theta^{\prime }+2kl^{2}\theta =Gl_{z}+T_{\theta }$$
where θ is the swing angle of the limit frame; J is the rotational inertia of the axis system; c is the equivalent viscous damping; l is the vertical distance from the limit seat to the shaft center; G is the weight of the load; $l_{z}$ is the vertical distance from the center of gravity of the load to the shaft center, and $T_{\theta }$ is the additional moment generated by the deformation of the axis system. Let u=θ, M=J, $C=2cl^{2}$, $K=2kl^{2}$, and $F_{G}=Gl_{z}+T_{\theta }$. Bring the appeal replacement into Formula (2); the small rotation can be equated with the translational motion shown in Figure 5:(3)$$Mu^{\prime \prime }(t)+Cu^{\prime }(t)+Ku(t)=F_{G}$$

Formula (3) is a typical second-order vibration system. When the system is overdamped or critically damped, the motion decays exponentially and does not oscillate, gradually returning to the original state after one equilibrium position at most. When the system is underdamped, it oscillates with decreasing amplitude at the equilibrium position. The structural parameters are presented in Table 3.

The equivalent mass of the axis-system mechanism is M=0.08 g, and the equivalent spring elasticity coefficient K=15.885 N/M. The critical damping of the system can be calculated as $c_{c}=2\sqrt{mk}=0.0713$ N/(m/s). From the MATLAB simulation, the velocity is taken as $\theta^{\prime }=0.4$ rad/s, and the additional torques generated by considering the load eccentricity without considering the deformation of the axis system are taken as $F_{G}=7\times 10^{-4}$ N. The results with different damping ratios are shown in Figure 6. The response time *t* occurs when the difference between the current position and the steady-state position is less than 1%. It is observed in Figure 5 that the system generates a positioning error due to the eccentric load.

If the damping of the system is too high, it hinders the system response; if the damping is too low, the system generates periodic oscillations, which deteriorates the dynamic performance [35,36,37]. From Figure 6, it can be seen that when the system damping ratio is 0.4, the system experiences significant oscillations and reaches a steady state after 34.9 ms. When the damping ratio is 1.3, the damping delays the system response, and the system reaches a steady state after 30.1 ms. When the damping ratio is equal to 1 or 0.7, the system response time is significantly reduced. Thus, the damping has a great influence on the response time of the system. The damping ratio of around 0.7 can better balance the system response speed and stability and is easier to achieve.

## 4. Experimental Analysis

To verify the response characteristics of the axis system structure in the optoelectronic-tracking platform, the experimental system shown in Figure 7 was built. The drive system used an STM32F7 series chip from ST (STMicroelectronics) as the core processor to control the motor velocity using a motor driver and collect the encoder data. The encoder (EAC58F21S18B) comes from the YIBEIJI company, with a repeatable position of less than 3 arc-seconds. The driver was developed by Nanjing University of Aeronautics and Astronautics. The bearings and other moving parts were well-lubricated. As the limit mechanisms of the pitch and azimuth shafts were similar, to simplify the experiment, only the motion characteristics of the pitch shaft were considered.

### 4.1. Response Characteristics at Limit Position

The response characteristics of the pitch-axis system limit mechanism at the limit position were tested. The limit holder was placed on the limit position from the initial alignment position under external torque, and the external torque was withdrawn, as shown in Figure 8a.

The encoder was sampled at 20 kHz. Five experiments were conducted, and the average value was calculated. The dynamic response of the axis system is shown in Figure 8b. It is observed that the mechanism is similar to that in an overdamped state. The limit holder can return to the initial position (148.26°) from the extreme position (150.41°) in 55 ms, with an error of <0.1°. There is almost no oscillation or overshoot; in actual operation, the limit holder is far from reaching the limit position.

### 4.2. Response Characteristics at Different Velocities

#### 4.2.1. The Complete Response of the Shaft

When the shaft turns to the target position at a constant velocity (fluctuation < 10%), the ultrasonic motor is powered off immediately. The power-off response time of the USM is within 2 ms. Therefore, the power-off position is the target position, and the automatic aligning time of the limit mechanism is the positioning time.

The defined shafting overshoot was the maximum value of the encoder reading angle minus the average of the last 1000 angle values after the USM was turned off. The defined positioning error was the angle value at power-off minus the average of the last 1000 angle values. The initial position was defined as the location where the visual alignment axis was perpendicular to the plane formed by the pitch and azimuth axes.

The USM drove the pitch shaft to move at uniform velocities of 30°/s and 230°/s for five revolutions back to the initial position before powering off. The response results during positioning are shown in Figure 9. The average axis-system overshoot and positioning error were determined after five sets of experiments.

When the shaft turns to the target position (186.09°) at a speed of 30°/s, the ultrasonic motor is immediately powered off. Due to the inertia of the axis system, the shaft crosses the target position. Under the action of the limiter, after about 5 s, the shaft stops at the limit position (186.23°), then after a small shock, about 7 ms, it returns to the target position with an error of 0.004°, as shown in Figure 9a. When the shaft is rotating at high speed (230°/s), the trend shown in Figure 9b is similar to that at low speed. Both the overshoot and error of limit mechanism alignment are increased, but the positioning time increased by only 13 ms. The positioning is extremely fast and with good positioning accuracy.

#### 4.2.2. Shaft Overshoot and Positioning Error

Due to the accuracy error of the axis system, the radial runout of the axis system varied at different positions, leading to changes in the structural forces. Therefore, the shaft should be experimented at each target position (0°, 90°, 180°, 270°).

After the USM drive axis system had been running steadily for two rotations at four different velocities, it was powered off. The pitch-axis overshoot at different positions after power-off is shown in Figure 10. The velocity is positively correlated with the amount of overshoot, which varies at different positions.

In the measurement, the radial runout of the pitch shaft had extreme values near 90°. The resistance torque was greatest at this position, resulting in position overshoots at 90° and 270° that were much smaller than those at 0° and its mirror position.

All errors take the absolute value. The positioning error of the pitch shaft is shown in Figure 11. The maximum positioning error $\varepsilon_{\sigma }\leq 0.12{}^{\circ}$. The variation in torque resistance at different positions of the axis system results in noticeable changes in positioning error with respect to position. In addition, due to the eccentric moment of the load, the positioning error of the pitch shaft at 0° is less than the positioning error at 180°, which is consistent with the simulation results in Figure 6.

At velocities less than 70°/s, the positioning error increases with velocity; the maximum positioning error $\varepsilon_{\sigma }\leq 0.05{}^{\circ}$. At velocities between 100°/s and 300°/s, the positioning error has no obvious impact on velocity but has an obvious correlation with the position. The positioning errors at 90° and 270° are much larger than those at 0° and 180°. It can be concluded that the resistance torque is greatest at the 90° position, which is consistent with our inference. In practical applications, different error compensations can be established by considering the speed and angle to improve the accuracy of the optoelectronic tracking platform.

Table 4 shows the comparison between the proposed optoelectronic tracking platform and other rotary equipment previously reported or available in the market. It can be seen that the positioning time of the proposed platform in this article is lower than most rotating devices, and positioning accuracy can also reach a higher level. Zhang has developed a two-axis non-magnetic turntable based on the ultrasonic motor, which has high positioning accuracy. However, during positioning, the shafting overshoot occurred, resulting in a much longer positioning time [34].

## 5. Summary and Discussion

In this study, a tracking platform driven by USMs is proposed. To reduce the size and mass of the drive system, the ultrasonic motor was integrated with the rotary shaft, and a novel limit mechanism was designed based on the self-locking feature of the ultrasonic motor. The limit mechanism was simplified and modeled, and the motion characteristics of the pitch-axis system were analyzed using theory and simulation. Some positioning experiments were conducted on the pitch-axis system to evaluate the performance of the limiter. The experimental results were consistent with the theoretical simulation analysis. The results indicate that the structure can effectively reduce the shock of USM. The error of automatic alignment is not only related to the speed but also to the precision of shafting under the action of the limiter. The maximum positioning error of the shafting system is less than 0.112°, and the maximum response time is less than 55 ms. When the operating velocity is less than 70°/s, the positioning error is less than 0.055°. Thus, the limiter structure can fully utilize the self-locking feature of the USM, enabling rapid automatic alignment of the axis system while maintaining excellent precision. This platform is suitable for scenarios that require rapid response without the need for extreme precision, and it has certain prospects for use. In the future, we will consider designing a more precise axis system to reduce the impact of positioning accuracy while also improving the limit mechanism to reduce friction torque and enhance the precision of automatic alignment.

## Figures and Tables

**Figure 1 micromachines-14-02067-f001:**
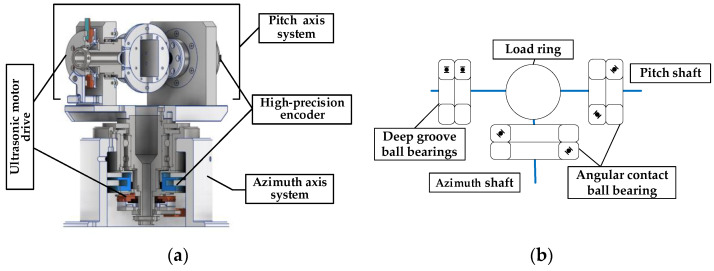
Overall structure. (**a**) Cutaway view of tracking platform. (**b**) Installation of bearing.

**Figure 2 micromachines-14-02067-f002:**
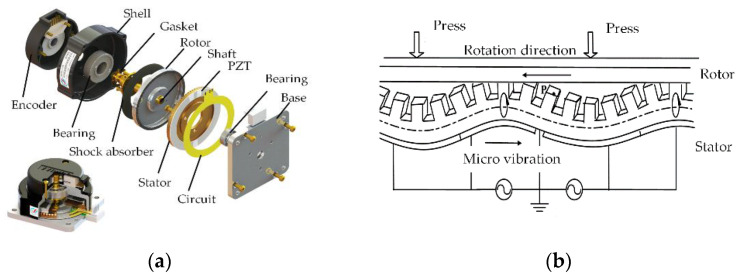
Traveling wave USM. (**a**) USM of pitch shaft. (**b**) USM of azimuth shaft.

**Figure 3 micromachines-14-02067-f003:**
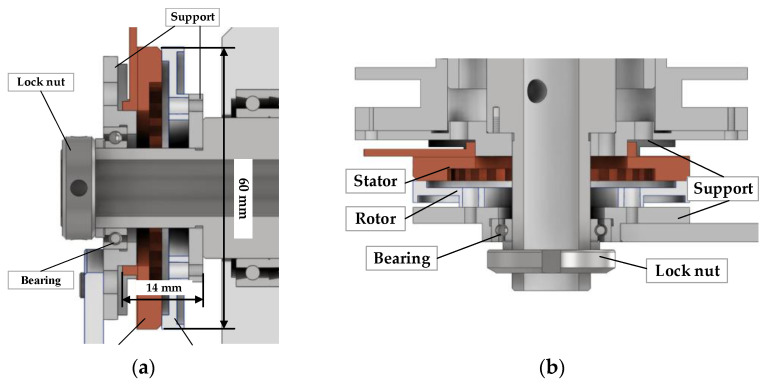
USM installation. (**a**) USM of pitch shaft. (**b**) USM of azimuth shaft.

**Figure 4 micromachines-14-02067-f004:**
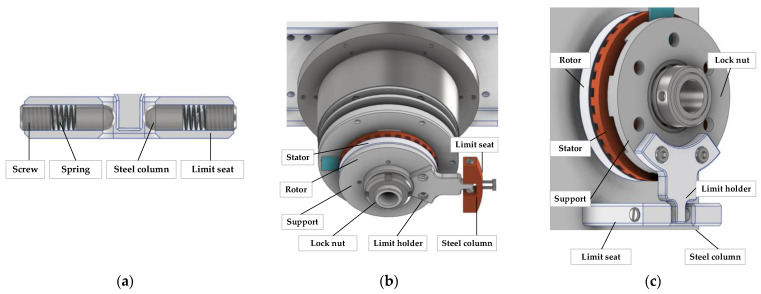
Mechanical angle limiter. (**a**) Limiter structure. (**b**) Limiter of azimuth shaft. (**c**) Limiter of pitch shaft.

**Figure 5 micromachines-14-02067-f005:**
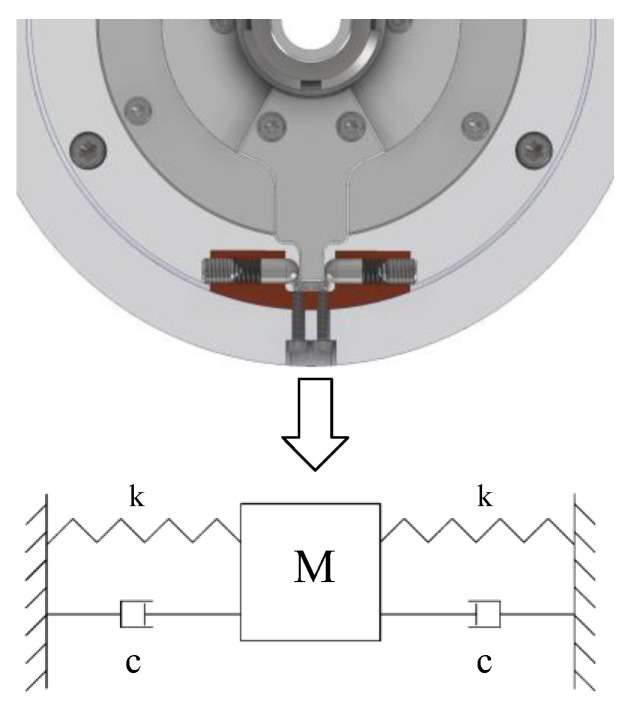
Principle of limiter structure.

**Figure 6 micromachines-14-02067-f006:**
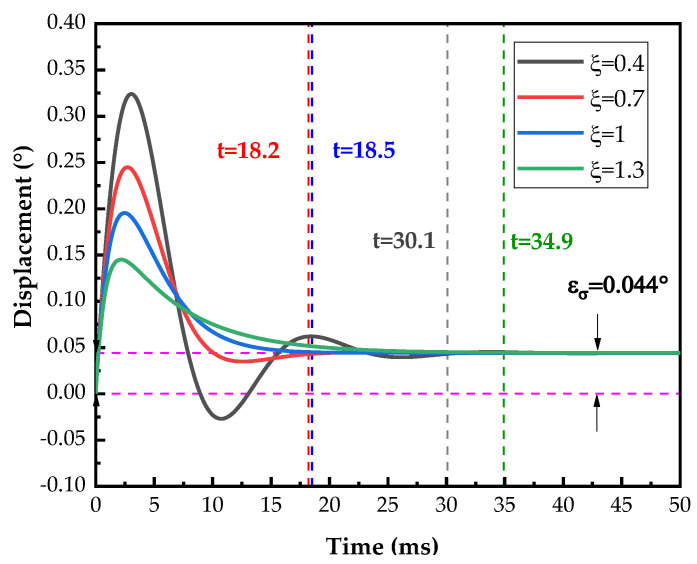
Overshoot of pitch shaft with different damping under load eccentricity.

**Figure 7 micromachines-14-02067-f007:**
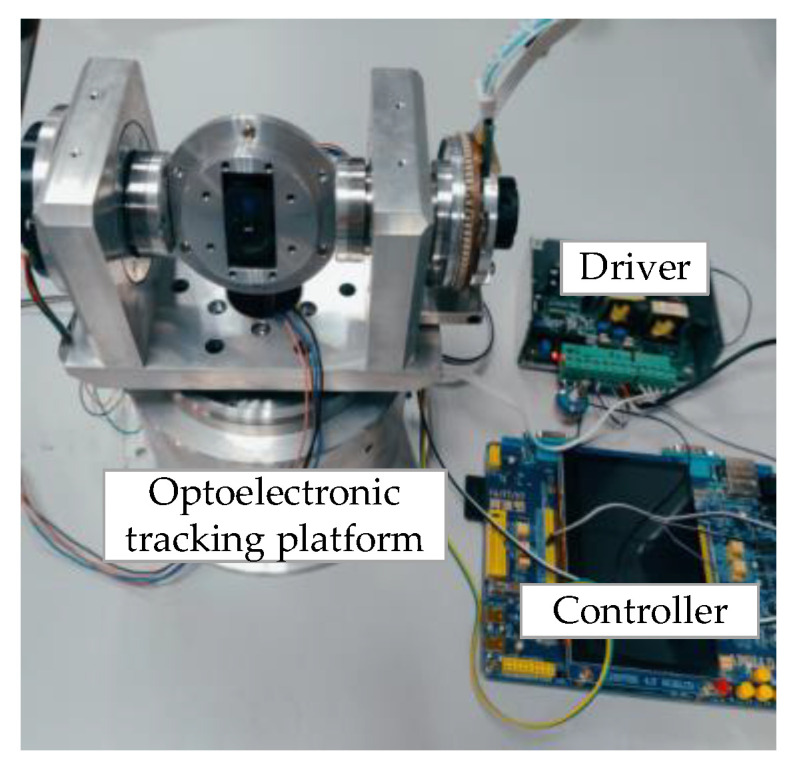
Optoelectronic tracking experimental platform.

**Figure 8 micromachines-14-02067-f008:**
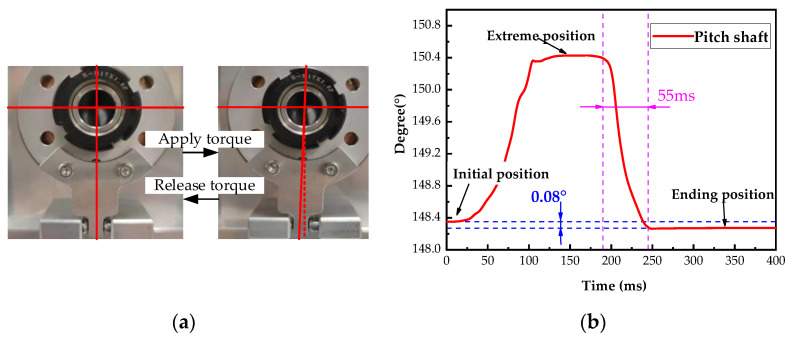
Pitch shaft response with limit mechanism at limit position. (**a**) Limiter structure experiment. (**b**) Degrees of pitch shaft change.

**Figure 9 micromachines-14-02067-f009:**
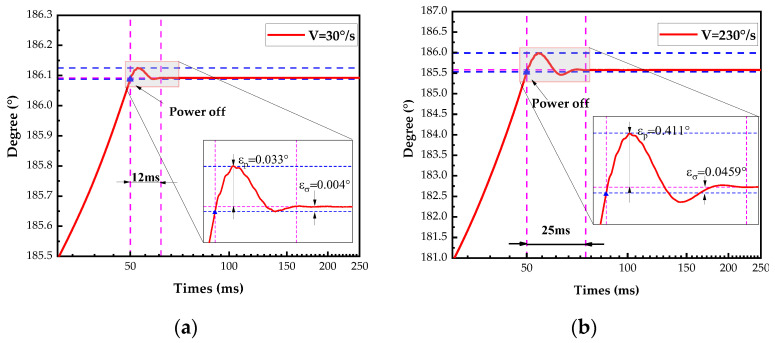
Motion of pitch shaft with power off. (**a**) Velocity of 30°/s. (**b**) Velocity of 230°/s.

**Figure 10 micromachines-14-02067-f010:**
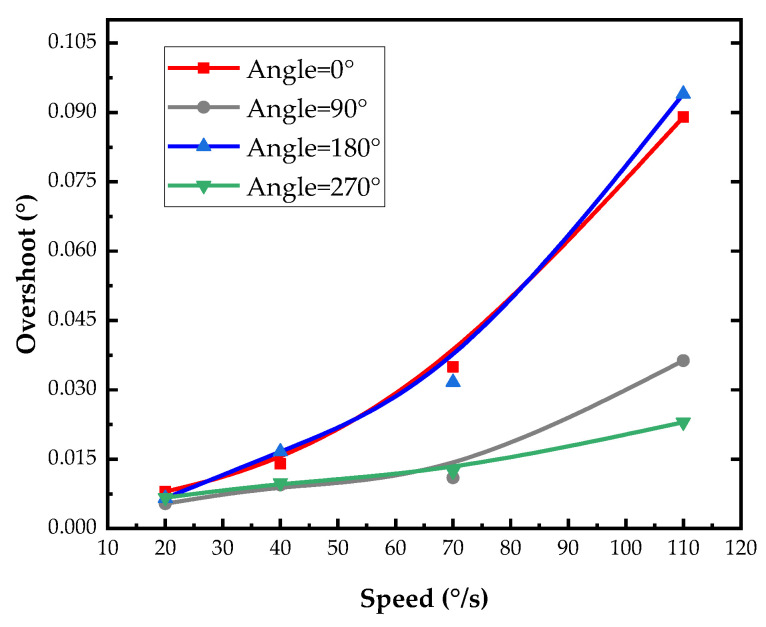
Pitch shaft overshoot at different velocities and positions.

**Figure 11 micromachines-14-02067-f011:**
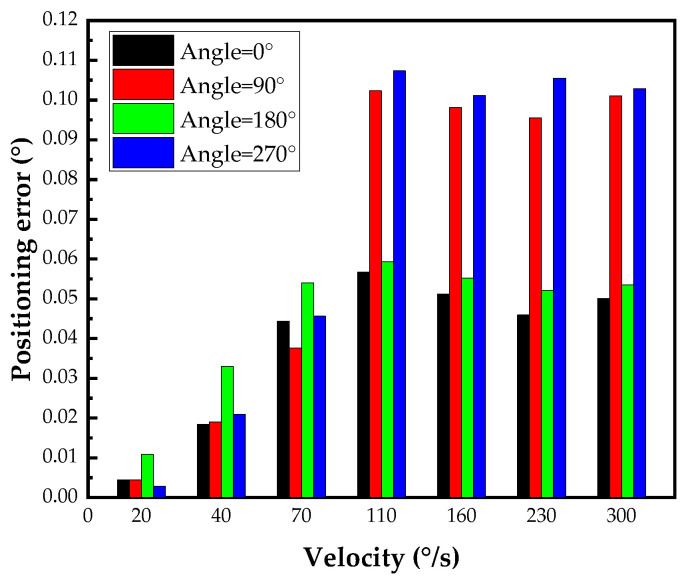
Pitch shaft positioning error at different velocities and positions.

**Table 1 micromachines-14-02067-t001:** The comparison between electromagnetic and ultrasonic motors.

Motor Classification	Manufacturers	Stall Torque/ (N/m)	Speed without Load/(r/min)	Weight/ g	Maximum Efficiency/%	Self-Locking
EM, DC, Brush	Maxon	0.0127	5200	38	70	No
EM, DC, Brush	Aeroflcx	0.00988	4000	256	20	No
EM, Alternating voltage/current, Three phases	Astro	0.0755	11,500	340	20	No
EM, DC, Brushless	Akribis	1.95	1500	350	80	No
USM, Traveling wave type, ∅60	PDLab	1.2	180	250	30	Yes
USM, Traveling wave type, ∅60	Proposed USM	1.5	180	285	30	Yes

**Table 2 micromachines-14-02067-t002:** Design parameters of optoelectronic tracking platform.

Parameter	Value
Size	174 mm × 208 mm × 170 mm
Azimuth angle	360° × N
Pitch angle	−80°/s~80°/s
Positioning accuracy	<0.02°
Positioning time	<2 s
Angular velocity	0.005°/s~180°/s
Angular acceleration	>120°/s^2^
Self-locking torque	>2.5 Nm

**Table 3 micromachines-14-02067-t003:** Structural parameters of pitch-axis system.

Parameter	Description	Value
*J* (kg·m^2^)	Rotational inertia	8 × 10^−5^
*l* (mm)	Length of limit holder	45
*k* (N·mm)	Elasticity of spring	3.9
*l_z_* (mm)	Eccentric gravity center	0.75
*G* (g)	Weight of load	93

**Table 4 micromachines-14-02067-t004:** Comparison of other rotary equipment.

Author	Motor	Positioning Accuracy	Positioning Time	Self-Locking Torque
Proposed platform	USM	<0.02° (open-loop, velocities < 30°/s)	<12 ms	>2.5 Nm
Bo Zhang [34]	USM	<0.00017° (closed-loop)	0.5 s~2 s	>4.4 Nm
Zhaolong Wu [17]	PMSM	<0.057°(closed-loop)	<15 ms (tracking unit step)	-
Xueyan Hu [18]	PMSM	0.0542° (closed-loop)	-	-

## Data Availability

The data presented in this study are available on request from the corresponding author.

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
