# Peer review of "Design of Optoelectronic Tracking Platform Driven by Ultrasonic Motor with a Novel Limiter"

_micromachines, 2023, doi:10.3390/mi14112067_

Round 1

Reviewer 1 Report

Comments and Suggestions for Authors

The authors have built an optoelectronic tracking platform using an ultrasonic motor instead of a conventional electromagnetic one.

Formal comments:
The structure of the article is roughly in line with the requirements for the structure of scientific articles. The language is clear and scientific. English is generally adequate, but a revision would be necessary, free of spelling errors and typos. The figures are of good quality and contain the information needed for understanding.

Substantive comments:
The introduction lists some of the disadvantages of electromagnetic motors to contrast with some of the advantages of ultrasonic motors, however there is not a compre-hensive comparison of the two types of motors, this is much needed in the article. It would also be good to have a figure to show the construction and operation of the two types of motors.

The introduction completely lacks any comparison of the new optoelectronic tracking platform presented with similar systems currently in use or in other research, so the novelty and significance of the research cannot be judged.

Session 2 starts with a detailed presentation of the platform created by the authors. In the introduction, the authors mention a number of uses, but in section 2 it is not made clear which of these uses their tool is intended for. It should be shown what tracking platforms are usually like, what sizes they are usually made in, what types they are, what are their main features, where their platform is positioned in relation to these, and why it was made with the parameters it was made with.

Section 4 presents in detail the results of the analysis of the device produced. It lacks any comparison with other devices, such as results from publications in the literature, results from systems using electromagnetic motors, results from other ultrasonic motor devices.

To summarise:
Although the section describing the authors' own work is fine, the presentation of the field of science and similar work, as well as the comparison of results with other results, is completely lacking, and therefore the value of the work cannot be judged and is not suitable for publication.

Comments on the Quality of English Language

"At present, the optoelectronic tracking platforms is driven by electromagnetic motor, which has problems of poor electromagnetic compatibility, big volume and small positioning stiffness[13-16]. "

singular - plural problems

Reviewer 2 Report

Comments and Suggestions for Authors

In this article, the authors propose a design of optoelectronic tracking platform with the limiter, which is driven by ultrasonic motors. This tracking platform can achieve automatic alignment of the axis system after ultrasonic motor is power-off and relieve the motor overshoot. Therefore, this design will reduce damage to optoelectronic equipment and precision shaft systems. This article can be published in the Micromachines after minor revision. Some comments are listed below.

1.    In line 83, the authors mention that the installation method can “effectively improve the rotary motion accuracy and extend the life of the precision rotary table”. This description is too vague and we cannot see any evidence to support this claim, such as some citations or experimental analysis.

2.    In line 167, the authors mention “If the damping of the system is too high, it hinders the system response” and you should cite the article which make the point.

3.    In line 192, the authors mention “Multiple experiments were conducted and the average value was calculated”. The authors should write clearly how many experiments were carried out.

4.    In line 194, the initial position is 148.26°and the extreme position is 150.41°. Why do the authors choose these degrees? Is it unreasonable for these angles to differ slightly?

5.    In figure5, figure7(b), figure8, figure9, and figure10, the upper and right axes should not have scales, just a straight line is good.

6.    The total number of citations in this article is too small, just only 21 citations, and it is reasonable to cite about 40 articles.

Comments on the Quality of English Language

None

Reviewer 3 Report

Comments and Suggestions for Authors

 The rotating traveling wave ultrasonic motor (USM) is a kind of line-of-sight (LOS) control mechanics. This manuscript develops an angel limiter for the ultrasonic motor to reduce braking torque influence when the motor stops. Although the response time, operating velocity, and positioning error of USM is not very suitable for the optoelectronic tracking with wide range, high speed, and high resolution, this manuscript is a good trying for find a new line-of-sight control mechanics.

My suggestions are following.

1 I suppose the Eq. (8) should be revised as Eq. (3). More explanations are needed on the relationships of Eq. (1), Eq. (2), and Eq. (3).

2 Why the damping ratio with 0.7 is the best choice? More explanations are needed about Fig. 5.

3 Is there any control algorithm design in the experiment to guarantee the best performance?

Comments on the Quality of English Language

Minor editing of English language required

Round 2

Reviewer 1 Report

Comments and Suggestions for Authors

The authors have corrected several serious problems with the first version of the article, but problems remain.

A figure showing the ultrasonic motor and an accompanying explanation have been included. It would have been nice to have seen an electromagnetic motor next to it, but even in this form the clarity has improved a lot, so this part is acceptable.

In section 4, a comparison with other similar systems has been included, which now allows the present publication to be judged, so that the results section is also acceptable.

The authors refer to the remaining problems in their reply, but they have not actually corrected the errors reported in the review. Perhaps they misunderstood the review.

It remains a problem that the introduction only highlights some of the disadvantages of electromagnetic motors and some of the advantages of ultrasonic motors, but there is no detailed, comprehensive, objective comparison to judge which type of device is best for what.

And although a comparison with other tracking platforms is included in the results section, the introduction completely lacks any comparison of the new optoelectronic tracking platform presented with similar systems currently in use or in other research, so the novelty and significance of the research cannot be judged.

Appreciating the corrections and trusting that the authors will provide the missing parts, I propose a minor revision.
